# Bones and Hormones: Interaction between Hormones of the Hypothalamus, Pituitary, Adipose Tissue and Bone

**DOI:** 10.3390/ijms24076840

**Published:** 2023-04-06

**Authors:** Olga Niwczyk, Monika Grymowicz, Aleksandra Szczęsnowicz, Marta Hajbos, Anna Kostrzak, Michał Budzik, Marzena Maciejewska-Jeske, Gregory Bala, Roman Smolarczyk, Błażej Męczekalski

**Affiliations:** 1Department of Gynecological Endocrinology, Poznan University of Medical Sciences, 60-535 Poznan, Poland; 2Department of Gynecological Endocrinology, Medical University of Warsaw, 00-315 Warsaw, Poland; 3Department of Cancer Prevention, Medical University of Warsaw, 02-091 Warsaw, Poland; 4UCD School of Medicine, University College Dublin, D04 V1W8 Dublin, Ireland

**Keywords:** hypothalamus, pituitary, fat tissue, bone, pituitary–bone axis

## Abstract

The bony skeleton, as a structural foundation for the human body, is essential in providing mechanical function and movement. The human skeleton is a highly specialized and dynamic organ that undergoes continuous remodeling as it adapts to the demands of its environment. Advances in research over the last decade have shone light on the various hormones that influence this process, modulating the metabolism and structural integrity of bone. More recently, novel and non-traditional functions of hypothalamic, pituitary, and adipose hormones and their effects on bone homeostasis have been proposed. This review highlights recent work on physiological bone remodeling and discusses our knowledge, as it currently stands, on the systemic interplay of factors regulating this interaction. In this review, we provide a summary of the literature on the relationship between bone physiology and hormones including kisspeptin, neuropeptide Y, follicle-stimulating hormone (FSH), prolactin (PRL), adrenocorticotropic hormone (ACTH), thyroid-stimulating hormone (TSH), growth hormone (GH), leptin, and adiponectin. The discovery and understanding of this new functionality unveils an entirely new layer of physiologic circuitry.

## 1. A Brief Overview of Bone Remodeling

Despite its apparent rigidity, the adult skeleton is a dynamic and metabolically active organ that undergoes constant renewal by remodeling [1]. The process by which bones undergo remodeling was first defined by American orthopedic surgeon Harold M. Frost in the 1960s. His theory proposed that bone is continually replaced and that damage is rectified instantly [2]. We now understand that remodeling is a dynamic process composed of four phases: activation, resorption, reverse, and formation [3]. The first phase, Activation, is initiated when bone lining cells (which thinly cover the external surface of bones) are activated and increase the surface expression of receptor activator for nuclear factor κ B ligand (RANKL). This activation can be triggered by any number of events, such as microfractures, hormonal changes, or stressors. The second phase, Resorption, is initiated with the activation of counterpart RANK, which is expressed on pre-osteoclasts and triggers osteoclast differentiation. Osteoclasts proceed with bone resorption and signaling to initiate osteoid matrix formation by osteoblasts. The third phase, Reversal, begins when new bone formation by osteoblasts overtakes bone resorption by osteoclasts. The last phase, Formation, completes the remodeling process, in which a new bone matrix is deposited by osteoblasts, mineralizing the collagenous osteoid matrix. Under physiological conditions, bone remodeling is a continuous cycle of bone resorption and reformation undertaken by osteoblasts and osteoclasts [4]. Osteoblasts arise from mesenchymal stem cells (MSCs) who themselves can differentiate numerous mesenchymal cell lineages. Osteocytes arise from osteoblasts that become encased in the bone matrix during the remodeling process and form an extensive network throughout bony structures. The functional role of the osteoblast is to produce a new bone matrix and promote matrix mineralization [5]. Osteoclasts arise from the monocyte/macrophage progenitor cell lineage and require RANKL activation for further differentiation and function [6].

Bone remodeling is an essential physiological process of skeletal change that begins in early fetal development and that continues throughout a person’s lifespan [7]. Throughout childhood and adolescence, bone size, mass, and composition are constantly changing. Assuring adequate peak bone mass (PBM) in the developing years is necessary to avoid osteoporosis in the future. A study by Xue et al. [8] found that the age at which peak BMD is achieved for the femoral neck, total hip, and lumbar spine was 18.7 years, 19.0 years, and 20.1 years, respectively, in females. The exact age at which BMD peaks varies by skeletal site and sex. The ultimate aim of the remodeling process is two-fold: to ensure functional bone adaptation to changing mechanic-skeletal needs, and to repair damage sustained by the bone matrix so as to prevent the accumulation of old, impaired bone [9,10]. The balance between these processes is crucial for sustaining bone homeostasis [11].

Bone remodeling is strictly controlled through a dynamic interplay of directly and indirectly influencing systemic and local factors, where any imbalance leads to a disturbance in bone remodeling. Systemic factors that affect remodeling include the parathyroid hormone (PTH), calcitriol, sex hormones, glucocorticoids, and thyroid hormones [12]. Local factors, on the other hand, include cytokines, prostaglandins, tumor growth factor-beta (TGF-β), and certain morphogenetic proteins. Other factors that affect bone mass and quality include nutrition, physical inactivity, chronic disease, and pharmacological agents. 

## 2. The Relationship between Hormones of the Hypothalamus and Bone

### 2.1. The Role of Kisspeptin on Puberty and the Reproductive System

Kisspeptin is a neuropeptide that is expressed by neurons of the hypothalamus. It acts as a key factor in regulating sexual development and function by moderating the pulsatile release of gonadotropin-releasing hormone (GnRH). Its main impact on reproductive function lies initially in regulating the onset of puberty and later in the maintenance of fertility [13].

The inactivation of mutations in the *KISS1* or *KISS1R* genes (which encode for kisspeptin and kisspeptin receptor, respectively) is found to cause the absence of puberty onset, whereas the activation of mutations of KISS1R causes precocious puberty [14]. These findings have cemented Kisspeptin as a crucial factor in initiating sexual maturation and maintaining reproductive health.

GnRH release is modulated primarily in the arcuate nucleus, where it is dependent on kisspeptin that acts in concert with neurokinin B and dynorphin [13]. Together, these neuropeptides regulate the function of the endocrine reproductive axis. Kisspeptin neurons located in the hypothalamus are also subject to metabolic regulatory agents, such as insulin, leptin, and ghrelin [15]. This strongly links metabolic balance to reproductive health. In turn, GnRH acts upon the endocrine axis spanning between the hypothalamus, pituitary, and gonads, initiating the release of luteinizing hormone (LH) and follicle stimulating hormone (FSH). Kisspeptin itself also has the potential to stimulate the surge of LH directly through neurons originating in the anteroventral periventricular nucleus [15]. They are also involved in the positive feedback control system that regulates steroid hormone levels.

### 2.2. The Influence of Kisspeptin on Bones

It is hypothesized that Kisspeptin has a positive metabolic effect on bone physiology. When considering its effect on skeletal development, it has been observed that a loss of function mutation in one of the kisspeptin genes causes a delay in skeletal maturation [16]. In contrast, Kisspeptin-activating mutations have been shown to cause the upregulation of skeletal maturation and remodeling [17]. Although the mechanism behind these observations can plausibly be attributed to a decrease or increase in sex steroid activity, we must consider equally the potential of kisspeptin as a neural regulator of bone metabolism. Numerous in vitro and in vivo studies have been undertaken to test these hypotheses. Kisspeptin-1 receptor (KISS1R) protein has been found highly expressed in MG-63 osteoblast-like osteosarcoma cells [18]. It is important to note that KISS1R is found expressed in human osteoprogenitor and skeletal stem cells [19].

Interestingly, Herber et al. [20] found that ovariectomized mice who underwent a selective ablation of estradiol receptor alpha in the arcuate nucleus went on to develop a 50% improvement in bone mineral density. This observation suggests that even in the absence of sex steroids, the kisspeptin circuit remains partially active, with a likely positive effect on bones. This can be regarded as evidence for the function of the neuro-skeletal axis.

Comninos et al. [21] studied the effects of kisspeptin on human bone metabolism in vitro and in vivo. In vivo, they demonstrated, for the first time in humans, the promoting effects of kisspeptin on the osteogenic differentiation of osteoblast progenitors, and, in vitro, the inhibition of bone resorption. Additionally, Kisspeptin was noted to cause an increase in osteocalcin (a bone formation marker), but no equivalent increase in the markers of bone resorption. This work highlights how kisspeptin acts directly on promoting the development of mature osteoblasts (osteoblastogenesis), rather than on increasing the activity of existing, mature osteoblasts. Crucially, the authors identified kisspeptin receptors on human osteoclasts and determined that the inhibition of osteoclast activity was dose-dependent. This suggests the possible anti-resorptive function of kisspeptin.

Further studies should focus on the mechanisms by which kisspeptin directly affects bone metabolism and should further define strategies for kisspeptin administration (dose and route of administration) for use as a potential therapeutic tool in osteoporosis management. 

### 2.3. Neuropeptide Y as an Important Hypothalamic Neurohormone

Neuropeptide Y (NPY) is a peptide made up of 36 amino acids that primarily acts in the central nervous system (CNS). It is active in many regions of the brain, including the arcuate and paraventricular nuclei of the hypothalamus. Beyond the CNS, NPY is found in the sympathetic nervous system and in the retinal pigment epithelium, smooth muscle tissue, intestinal tissue, and bone marrow [22].

Neuropeptide Y functions mainly as a co-transmitter to sympathetic neurotransmitters. It binds to specific Y receptors that inhibit calcium and activate potassium channels [23]. Five different types of Y receptors have been identified so far. Both the variety and abundance of NPY in the human organism suggest its wide range of roles and extensive involvement in the physiological processes of the body. NPY expression and transmission is involved in the regulation of blood pressure, sexual behaviors, the function of immune cells, or the processing of fear, anxiety, and memory. It also plays a significant role in the proliferation of cells in the retina, as well as in the neuromodulation and neuroprotection of retinal structures.

NPY plays a significant role in maintaining metabolic balance. Metabolic outcome, however, is contingent on the type of Y receptor activated, where NPY can either promote the intake of energy or inhibit feeding mechanisms [24]. NPY is also an integrated conduit for signals generated by peripherally produced hormones such as insulin and leptin. Neuropeptide Y neurons have been found to express receptors that are specific to anorexigenic hormones. Upon activation of these receptors, the expression of NPY is down-regulated to decrease appetite [24].

Further research into the function of NPY could therefore be beneficial in the treatment of a variety of ailments, such as diabetic retinopathy, glaucoma, post-traumatic stress syndromes, or metabolic syndrome [25].

### 2.4. The Influence of Neuropeptide Y on Bones

Recent advances in the study of neuropeptide Y have provided interesting insight into novel neuronal pathways that join the brain and skeletal system. In parallel with Kisspeptin, Neuropeptide Y is just as important a neuropeptide with regard to its impact on bone [26]. Its influence is multifactorial and includes direct and indirect mechanisms. 

NPY and its relevant receptors have been identified in bone tissue and there is evidence to suggest that NPY modulates local bone remodeling [27]. It is suspected that NPY operates in a mediatory capacity between the autonomic nervous system and bone marrow mesenchymal cell differentiation. Zhang et al. [28] confirmed that administering NPY in mice decreased the expression of runt-related transcription factor 2 (runx2) during the osteogenic differentiation of bone marrow stromal cells (BMSCs). NPY has been observed to play a role in the process of bone resorption, however results from these studies remain inconclusive [10,29]. Baldock et al. [30] have shown that NPY signaling is integral to the regulation of bone mass as it relates to the centrally perceived energy status of the body.

The indirect action of NPY on bone is thought to extend from its influence on the gut microbiota and blood vessel formation. As such, it is largely perceived as an intermediary in the autonomic nervous regulation of bone metabolism [31].

Further research into the effects of NPY on bone effect is essential, not only to establish physiological effects, but also in order to develop novel treatments for bone diseases.

### 2.5. Role of Neuromedin U in Bone Metabolism

Neuromedin U (NMU) was first discovered in 1985 from porcine spinal cord isolates, while its specific receptors, neuromedin U receptor 1 (NMUR1) and neuromedin U receptor 2 (NMUR2), were subsequently identified in the early 2000s. [32]. Since their discovery, multiple studies have demonstrated their pleiotropic effects on the human body. It has been shown that NMU participates in a wide range of physiological and pathophysiological processes, including the contraction of smooth muscles, energy balance, feeding behaviors, stress responses, and inflammatory responses [33,34,35]. NMU is an anorexigenic neuropeptide that acts independently of leptin but whose mechanisms of action remain poorly defined. Numerous studies that postulate the potential role of NMU in bone remodeling have been published. A pioneering animal study by Sato et al. [36] has reported that a global deficiency of NMU in Nmu mutant mice leads to increased bone mass and is associated with increased osteoblast activity. This discovery has laid the groundwork for further investigation, which continues into the present day. No definitive conclusion has been established, however, as to whether NMU controls bone formation directly or indirectly. Hsio et al. [37] recently corroborated the a priori findings published by Sato, and also demonstrated that the administration of synthetic NMU restricts osteoblastic differentiation. These findings may significantly contribute to the development of novel therapeutic agents.

### 2.6. The Role of Cocaine- and Amphetamine-Regulated Transcript in Bone Metabolism

Cocaine- and amphetamine-regulated transcript (CART) is an anorexigenic neuropeptide that is widely expressed in the central nervous system and in peripheral tissues, yet no specific receptor for these peptides has been clearly defined [38]. Besides its involvement in feeding behavior and stress responses, the hypothalamic function of CART is largely unknown [39]. CART is one of two known mediators involved in the regulation of bone mass by leptin, where it inhibits bone resorption through the modulation of RANKL expression [40]. Experimental animal models by Singh et al. [41] have shown that Cart(^−/−^) mice develop a low bone mass phenotype due to an isolated increase in osteoclast number. Additionally, a mere two-fold increase in circulating CART levels was enough to produce an increase in bone mass due to an isolated decrease in the osteoclast number [41]. Despite these findings, the mechanism by which CART regulates bone resorption under the control of leptin is still not well defined and poses a challenge for future research.

### 2.7. The Role of the Endocannabinoid System in Bone Metabolism

The endocannabinoid system plays a significant role in regulating numerous physiological processes, including neurotransmission, appetite control and energy balance, pain perception, and immune responses. Endocanabinoids are believed to be involved in bone remodeling as they and their receptors, cannabinoid receptor type 1 (CB1) and cannabinoid receptor type 2 (CB2), have been identified in elements of the skeletal system [42]. Idris et al. [42] have reported that CB1 regulates the process of age-related osteoporosis, a process that is characterized by reduced bone formation and the accumulation of adipocytes in the bone marrow compartment. Furthermore, they demonstrated in an animal model that mice with CB1 deficiency developed a higher peak bone mass due to a decrease in bone resorption, yet still developed age-related osteoporosis [42]. This study has provided evidence that CB1 receptors have a dual function; on one hand, they regulate peak bone mass by affecting osteoclast activity, whereas on the other hand, they protect against age-related bone loss by regulating the adipocyte and osteoblast differentiation of bone marrow stromal cells. In a study by Tam et al. [43], the bone formation rate was reduced in young CB1-deficient mice, confirming that CB1 plays a crucial role in regulating bone formation and achieving peak bone mass. These findings all demonstrate the therapeutic potential of cannabinoid receptor ligands in enhancing peak bone mass and in preventing age-related osteoporosis. Cannabinoid receptors should be explored further for the possible development of new treatments for bone diseases in the future.

## 3. The Possible Role of Pituitary Hormones on Bone Metabolism: FSH, PRL, ACTH, TSH, GH as a Pituitary–Bone Connection

For years, the role played by pituitary hormones in bone metabolism was viewed only in terms of their indirect effects through peripheral endocrine organ mediators. More recently, studies have shown that despite their mediated effects, pituitary hormones may also exert a direct effect on the skeleton. This novel area of research, focusing on the pituitary–bone axis, uncovers new possibilities for the diagnosis and treatment of metabolic bone disease.

### 3.1. The Role of FSH in Bone Metabolism

In 2006, Li Sun et al. [44] conducted a pioneering study to explore claims that the follicle-stimulating hormone affects bone health directly and in an estrogen-independent pattern. They demonstrated that FSH directly impacts skeletal remodeling via the stimulation of osteoclast formation and function. FSH receptors (FSHR) were found on osteoclasts and mesenchymal stem cells, but not on osteoblasts [45]. Moreover, they observed that female mice lacking either FSH or the FSHR were resistant to bone loss, despite their hypogonadism [46]. In subsequent research that further built on these findings, Jie Wang et al. [46] reported a correlation between serum FSH and the occurrence of osteoporosis in postmenopausal women. This has led to speculation regarding the use of FSH as a predictor of bone loss in this group.

### 3.2. The Role of PRL in Bone Metabolism

For decades, hyperprolactinemia has been a well-known risk factor for the development of decreased bone mass. It was initially believed that prolactin-mediated hypogonadism, caused by the suppression of pulsatile GnRH release, was the underlying cause. This, however, was challenged by evidence that patients with prolactinomas (and in whom no hypogonadism had developed) exhibited an increased risk of bone fractures [47]. These observations provided evidence to suggest that increased PRL levels may affect bone metabolism directly. Prolactin receptors were found expressed on osteoblasts, while their biological response was observed to be PRL-concentration-dependent. Although a physiological PRL concentration is crucial for healthy bone homeostasis, at elevated levels, bone resorption overtakes bone formation. The correlation between PRL levels and the intensity of bone resorption is well documented. PRL levels that correspond to those observed physiologically during pregnancy or lactation (between 100 and 500 ng/mL) tend to promote bone resorption via the stimulation of osteoclastogenesis through RANKL upregulation. When higher, pathological PRL levels (i.e., 1000 ng/mL) manifest in the inhibition of osteoblastogenesis and bone formation [48].

### 3.3. The Role of ACTH in Bone Metabolism

Adrenocorticotropin (ACTH) is believed to influence bone metabolism via the melanocortin 2 receptor (MC2R), but studies on the exact effects of this interaction have so far proved inconclusive [49]. Zaidi et al. [50] report that ACTH prevents glucocorticoid-induced osteonecrosis, while in vitro ACTH administration stimulates the proliferation of osteoblasts in a dose-dependent manner. These findings suggest that ACTH can be considered a therapeutic target for treating osteonecrosis of the femoral head.

### 3.4. The Role of TSH in Bone Metabolism

Investigations into the direct effects of TSH on bone metabolism have been undertaken since the late 1990s, when Inoue et al. first reported the expression of TSH receptors (TSH-R) on the surface of osteoblasts [51]. It took a further decade of study to then demonstrate that osteoclasts possess functional TSH-R [52]. Until this point, osteoporosis that is associated with hyperthyroidism was viewed as a secondary consequence of altered thyroid function. Animal models were used to show beyond doubt that the reduced expression of TSH-R led to decreased bone density [53]. Li Sun et al. [54] further demonstrated the antiresorptive action of TSH when they observed improved bone microstructure and the prevention of osteoporosis following the administration of low-dose recombinant human TSH in ovariectomized rats.

### 3.5. The Role of GH in Bone Metabolism

The growth hormone (GH) and insulin-like growth factor-1 (IGF-1) axis has a pleiotropic effect on the skeleton, influencing bone formation and resorption [55]. Animal studies, however, have provided evidence to support that GH directly affects bone, bypassing IGF-1 entirely through a growth hormone receptor [56]. While research suggests that the skeletal effects of GH require IGF-1, there is growing evidence that GH can act independently of IGF. Fritton et al. [57] used a liver IGF-1-deficient (LID) mouse to show that GH protects against ovariectomy-induced bone loss in states of low circulating IGF-1. This observation is very relevant as it relates to states of low serum IGF-1 and estrogen deficiency (i.e., postmenopausal women), in which increases in GH levels could be used to protect against bone loss.

## 4. The Roles of Leptin and Adiponectin in Energy Homeostasis and Neuroendocrine Regulation

As research into adipocytes continues to evolve, it has emerged that adipose tissue does is not used exclusively for energy storage as was once believed, but that it is an important endocrine organ that can secrete numerous hormones and that controls a wide range of bodily functions. Adipocytes are responsible for producing adipokines, such as leptin, adiponectin, resistin, chimerin, visfatin, vaspin and apelin.

The discovery of leptin led to a fundamental shift in our understanding of energy balance and has driven a change in our perception of adipose tissue as an active endocrine organ. The majority of leptin production has been found to occur in adipose tissue, while levels of circulating leptin (a marker of energy status) correlate strongly with body fat content. In addition to its effect on homeostasis, neuroendocrine and immune system function, leptin also regulates glucose, lipid, and bone metabolism [58].

Leptin’s pulsatile secretion follows a circadian rhythm, with its lowest levels at midday and its highest at midnight. Although both lean and obese patients secrete leptin in a pulsatile rhythm, the amplitude of pulses in obese patients has been found to be much larger. Leptin levels fluctuate in line with changes in caloric intake, with a significant decrease occurring during starvation [59]. Despite its significant decrease after menopause, women have higher serum leptin levels at baseline than men do. This suggests gonadal hormone involvement and marked sexual dimorphism. In addition to sex-related hormones, insulin, glucocorticoids, catecholamines, and cytokines also exert an effect on leptin levels [60].

Leptin’s significance is best understood in the context of leptin deficiency. The complete absence of leptin results in severe hyperphagia, decreased metabolic rate, and rapidly developing obesity, which can be reversed with the subsequent administration of leptin. When leptin is supplemented in healthy subjects, calorie consumption, body weight, and body fat level decrease significantly.

In the most prevalent forms of obesity, particularly lifestyle-related cases in which overeating and a sedentary lifestyle are culprits, an insensitivity to spontaneous hyperleptinemia or leptin therapy presents as “leptin resistance”. Leptin regulates appetite by activating anorexigenic neurons while decreasing the activity of orexigenic neurons. During fasting, leptin levels in adipose tissue and plasma fall precipitously and increase energy expenditure via sympathetic nerve stimulation [61]. In a study by Calbet et al. [62], circulating leptin levels were found to decrease in obese men who achieved a negative energy balance through activity and calorie restriction, suggesting that leptin levels are indicative of energy status. Low leptin levels during fasting induce a host of metabolic and hormonal responses. These include hyperphagia, hypogonadotropic hypogonadism, and the suppression of thyroid and growth hormone release, all processes that are suppressed by leptin at physiologic levels [63]. Leptin plays a crucial role in reproduction, and as such, leptin replacement therapy is used to modulate pubertal development in leptin-deficient humans. Leptin delays the onset of puberty, lessening the effects of starvation-related pubertal delay. Hypogonadism, impaired leptin pulsatile secretion, and generalized lipoatrophy are associated with low leptin levels [64]. Treatment with leptin improves hypothalamic amenorrhea by normalizing thyroid and cortisol levels, increasing luteinizing hormone pulse frequency, and increasing estradiol levels. 

Patients with generalized lipodystrophy, who have low leptin levels and insulin resistance, respond well to leptin replacement, which normalizes LH and sex steroid levels. It is becoming increasingly apparent that leptin may play a crucial role in communicating information between energy reserves and the neuroendocrine axis.

Adiponectin is yet another adipokine produced by adipose tissue and that exerts neuroendocrine effects. Adiponectin has anti-inflammatory properties, improves insulin sensitivity in obesity, and has a positive impact on the endocrine system as a whole [65]. Moreover, adiponectin has been shown to induce apoptosis in malignant cells, and also has anti-oxidant and anti-inflammatory properties.

The benefits of adiponectin in preventing diseases of the central nervous system are due to its insulin-sensitizing, anti-inflammatory, angiogenic, and vasodilatory capabilities. Several crucial brain processes are believed to be directly influenced by adiponectin, including energy balance, neurogenesis in the hippocampus, and synaptic plasticity. It has been found to cross the blood brain barrier via peripheral circulation, has been shown to inhibit glial cell differentiation, and suppresses inflammation [66].

Plasma adiponectin concentration is inversely associated with weight, central obesity, the risk of type 2 diabetes, and insulin resistance. Adiponectin increases skeletal muscle glucose uptake, insulin sensitivity, and fatty acid oxidation [67]. Studies have shown that obesity, diabetes, atherosclerosis, and nonalcoholic fatty liver disease are all linked to lower levels of adiponectin. On the other hand, fasting and weight loss are both linked to higher serum adiponectin. The exact physiological processes by which these changes are associated with adiponectin are still being investigated. It is known that adiponectin has very strong anti-inflammatory, anti-apoptotic, anti-fibrotic, and proangiogenic effects. It is very likely that many physiological and pathophysiological effects of adiponectin are yet to be discovered [68].

### 4.1. Leptin Dependent Regulation of Skeletal Function: Direct and Indirect Mechanisms

Many endocrine factors originate from adipose tissue and play a role in metabolism and neuroendocrine regulation. Of these, leptin also appears to play a major role in bone metabolism. Leptin is an adipokine produced by adipocytes and secreted in a pulsatile fashion to maintain energy homeostasis. It has a direct anabolic effect on chondrocytes and osteoblasts; however, it also modulates bone physiology indirectly by acting on the hypothalamus, the sympathetic nervous system, and via changes in body weight [69].

The expression of leptin receptors is observed on osteoblasts, chondrocytes, as well as mesenchymal stem cells undergoing osteogenic differentiation, which suggests that leptin acts directly on skeletal physiology. Recent studies in which primary osteoblasts were cultured with leptin have demonstrated a dose-related increase in mineralized bone nodule formation. Leptin has also been shown to play a critical role in the differentiation of marrow stromal cell lines into osteoblasts [70]. Motyl et al. [71] demonstrated leptin-dependent increases in osteoblast proliferation. Gordeladze et al. [72] observed that leptin promotes the synthesis of type I collagen, bone matrix proteins, osteocalcin, and affects bone mineralization. Legiran et al. [73] further concluded that leptin may also play a significant role in the inhibition of bone resorption. They found that leptin decreases osteoclast differentiation and stimulates the expression of osteoprotegrin, inhibiting osteoclastogenesis. Previous studies had shown no effect of leptin on mature osteoclasts; thus, leptin appears to act on osteoblasts to regulate osteoclastogenesis, but does not act itself on mature bone-resorbing cells [7].

It has been shown that beyond acting on its own receptor, leptin may also influence bone growth directly through the activation of fibroblast growth factor-23 (FGF-23) [74].

Beyond acting locally on bone metabolism, there are several well-documented mechanisms of leptin regarding its indirect effects on bone physiology. Takeda et al. [75] were the first to describe leptin-regulating bone formation via the sympathetic nervous system (SNS). They observed that the ventromedial hypothalamus activated local noradrenergic signaling pathways to osteoblasts in response to leptin. Serotonin binds to serotonin 5-hydroxytryptamine receptor 2C (5-HT_2C_) in the ventromedial hypothalamus and serotonin 5-hydroxytryptamine receptor 1B (5-HT_1B_) on osteoblasts, causing the inhibition of bone growth. Leptin causes a decrease in neuronal serotonin synthesis, and also inhibits serotonergic receptors [76]. On the other hand, leptin deficiency has been shown to downregulate the transcription and expression of the neuropeptide CART (cocaine and amphetamine-regulated transcript) in the hypothalamus, which, in turn, promotes bone resorption by modulating RANKL expression on osteoblasts [40,69].

The balance of the central and peripheral effects of leptin on bone metabolism remains a strongly debated topic and an area of ongoing study. Leptin has direct anabolic effects on bone cells, and simultaneously indirectly influences bone metabolism. To date, most studies using animal models to explore these mechanisms have yielded conflicting and inconclusive results. 

Starting with the assumption that gonadal failure induces bone loss while obesity prevents it, Ducy et al. [77] studied leptin-deficient (ob/ob) and leptin receptor-deficient (db/db) mice. They observed that both mutant mouse lines demonstrated increased bone formation leading to high bone mass despite hypogonadism and hypercortisolism. The authors of the study concluded that the development of this phenotype is independent of the presence of fat, and is specific to the absence of leptin signaling. The authors identified leptin as a potent inhibitor of bone formation acting through the central nervous system. However, in similar analyses conducted by Williams et al. [78] and Turner et al. [79], bone formation was reduced in both ob/ob and db/db mice. These findings contradict the initial findings by Ducy et al., but are also confirmed in later studies. These latest results demonstrate a reduction in bone mass in the absence of leptin signaling, indicating that leptin acts in vivo as an anabolic bone factor and increases osteoblast number and activity. These findings suggest that the direct effects of leptin on bone cells may override its actions via the central nervous system.

Equally controversial is the role of leptin in the hypothalamus and in the process of bone formation. Ducy et al. [77] and Takeda et al. [75], as discussed above, demonstrated the overall negative effect of leptin acting in the hypothalamus, whereas Turner et al. [79] and Kalra et al. [80] have demonstrated the opposite. Although observations on the effect of leptin on the hypothalamus should not be discounted, this effect appears to be minimal compared to the direct anabolic effect of leptin on bone [80].

It is thought that leptin may also act through serotonergic signaling in the brainstem. As mentioned above, serotonin decreases bone growth by binding to its receptors in the hypothalamus and on osteoblasts [26]. Upadhyay et al. [81] have shown that while leptin inhibits the activity of serotonin in models both in vitro and in vivo, it, by extension, promotes bone formation.

As there is a significant overlap and a strong dependency between leptin and its complementary hypothalamic effectors, such as estrogen, cortisol, IGF-1, and parathyroid hormone, it is difficult to isolate the direct and indirect effects of leptin on bone metabolism [82]. These hormones are all acted upon by leptin, which also positively contributes to improved bone mass. Estrogen itself activates osteoblast differentiation, while cortisol is inhibited by leptin through the hypothalamic–pituitary–adrenal axis. Hypercortisolemia is a well-known contributor to the loss of bone density, while the inhibitory action of leptin on cortisol activity may in fact promote bone formation [83]. The multidirectional effect of leptin on bone metabolism is presented in Figure 1.

Recently, a strong indication of the positive association between leptin levels and bone growth, density, and structural integrity has emerged. Hyperleptinemia, on the other hand, may lead to a state of leptin resistance that has been seen primarily in the obese population as a reduced hypothalamic response to leptin. Recent studies show that bone mass is inversely related to the percent of fat mass [84,85]. At higher concentrations, leptin acts as a proinflammatory adipokine and may trigger inflammatory pathways in osteoblasts, causing a decrease in bone density and strength [86].

### 4.2. Adiponectin Dependent Regulation of Skeletal Function: Direct and Indirect Mechanisms

Adiponectin is yet another adipocytokine that influences bone metabolism. Recent studies have shown that adiponectin blocks the differentiation of bone marrow macrophages and mononuclear cells into osteoclasts by inhibiting the macrophage colony stimulating factor (M-CSF) and receptor activator for nuclear factor κ B ligand (RANKL). Through this process, it manages to suppress the bone resorption activity of these cell lines [44,87,88]. It was also found to increase the expression of alkaline phosphatase and promote the mineralization activity of osteoblasts. Overall, this promotes bone formation and limits resorption by activating osteoblastogenesis and suppressing osteoclastogenesis [89]. Adiponectin encourages the proliferation, survival, migration, and mineralization of osteoblasts, and simultaneously limits the proliferation, survival, and migration of osteoclasts [90].

## 5. Clinical Utility of New Factors in the Practice of Metabolic Bone Diseases

Advances that demonstrate the important role played by adipokines, as well as by reproductive hormones other than sex steroids, in bone physiology, provides hope for their potential future utility as a clinical treatment [91].

In 2006, Sun et al. [44] demonstrated that neither FSHβ or FSH receptor (FSHR)-null engineered mice demonstrated bone loss, despite marked estrogen deficiency. Moreover, observational studies in humans have shown that follicle-stimulating hormone levels, which increase gradually in the years preceding menopause, and marked estrogen depletion in perimenopausal women, correlate better with bone mass loss than levels of gonadal steroids [92,93]. These observations have paved the way for new hypotheses proposing that FSH has important extragonadal functions. Osteoclasts express FSHR, which suggests that bone resorption may be directly stimulated by the follicle-stimulating hormone. Studies in mice indicate that FSH may also regulate adiposity. Such findings have led to tentative attempts at using FSH antibodies to simultaneously treat age-related osteoporosis and obesity [94]. In a similar approach, polyclonal and monoclonal antibodies were found to increase bone mass in gonadectomized mice [95]. Humanized FSH antibodies will also profoundly inhibit the action of FSH in cell-based assays, a finding which may be prelude to further preclinical and clinical testing.

Critics are quick to point out, however, that experiments using global FHSβ or FSHR null mice are confounded by changes in LH and androgens. Direct interventional studies, in which FSH is either suppressed or infused, have not provided substantial evidence to the effect that FSH regulates bone turnover in humans independent of changes in sex steroid levels [96,97,98,99,100].

There exists abundant experimental evidence that suggests that human adipose tissue can influence bone metabolism through the biochemical mediation of adipokines, which also play a role in the pathophysiology of osteoporosis [101,102]. However, studies on how best to use this knowledge to address treatment targets are sparse. Circulating levels of adipokines in the serum or plasma may not be representative of the local processes influencing bone remodeling [103]. Therefore, these metrics may not serve as a reliable diagnostic, therapeutic, or monitoring marker. Several studies have emphasized that bone marrow adipocytes, rather than systemic adipose tissue, are the potential regulatory players that control the interaction between osteoblasts and osteoclasts during bone remodeling. 

In vitro studies in which the human adiponectin gene was transfected into adipose-derived stem cells via recombinant adenovirus and was co-cultured with bone marrow mesenchymal stem cells have shown the inhibition of adipogenesis and promotion of osteogenesis [104]. In a tibia fracture rat model, the local administration of human recombinant adiponectin increased bone healing [105]. These findings support the hypothesis that the local administration or local gene-transduction, rather than systemic administration, may provide better positive effects on bone formation.

The balance between the central and peripheral effects of leptin on bone remains an area of substantial controversy. Systemic leptin administration in animal studies has shown a positive effect on bone formation, except when administered in very high doses. A small number of human studies, in which recombinant human methionyl leptin was administered, resulted in the preservation of or increase in bone mass. These observations were particularly pronounced in leptin-deficient individuals [69,84]. These studies, however, do not have the resolution to determine whether the observed effect is a result of direct leptin action on bone or is mediated by changes in sex hormones, thyroid hormones, or IGF-1. There is an obvious need for a large, controlled, clinical trial in order to explore this therapeutic option in osteoporosis and other metabolic bone diseases.

Chemerin is another adipokine that is recognized as a marker of inflammation and metabolic syndrome. It is characteristically elevated in individuals with osteoporosis. Chemerin has been shown to boost mature osteoclast activity, while its action can be effectively inhibited by a chemokine-like receptor antagonist [106]. In an animal model, a chemerin receptor antagonist effectively reduced alveolar bone loss, a finding which would suggest that chemerin participates in periodontitis-induced bone loss. Additional research is required, however, to explore whether chemerin has the same effect on bone loss in long bones. Such findings would present the possibility of another potential target for this new line of treatment.

Beyond individual adipokines, additional lines of inquiry that may contribute to our understanding of the regulation of bone metabolism is the cross-talk between myokines (derived from myocytes IL6, irisin, BDNF, myostatin and FGF2), osteokines (osteocalcin and sclerostin derived from bone cells), and adipokines [107].

Both adipocytes and osteoblasts originate from the same stem cell line, the mesenchymal multipotent stem cells (MSCs). Therefore, cell therapy with MSCs may contribute to bone loss prevention. MSCs may differentiate into osteoblasts at one location, and secrete specific growth factors at another, together promoting bone formation. Recent studies have observed that extracellular vesicles (EVs) from MSCs possessed therapeutic potential for the treatment of osteoporosis, similar to that of progenitor cells [108]. Currently, multiple clinical trials are underway to study the effect of infusion cell therapy for the treatment of osteoporosis [109].

## 6. Conclusions

The regulation of bone formation and resorption is underpinned by the interaction of estrogens, testosterone, parathyroid hormone, vitamin D, thyroid gland hormones and glucocorticoids. There is a growing body of evidence, however, which argues that additional factors may be involved. Factors that have been traditionally connected with the nervous system, such as kisspeptin, or with adipose tissue, such as leptin and adiponectin, may be critical contributors to bone turnover and the pathogenesis of some metabolic bone diseases. In Table 1, we provide an overview of the most important hormones included in this review, with a summary of their origin, main functions, and bone metabolism-specific function. 

Emerging evidence of newer reproductive hormones and adipokines has amplified our understanding of the development of bone diseases and gives hope for the development of novel, safe, and effective therapies.

## Figures and Tables

**Figure 1 ijms-24-06840-f001:**
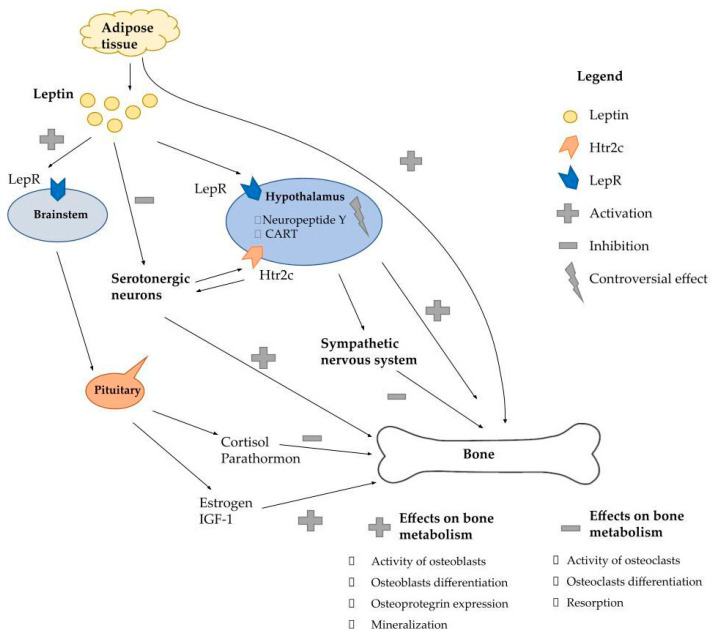
The multidirectional effect of leptin on bone metabolism. Lepr—leptin receptor; Htr2c—5-hydroxytryptamine receptor 2c; CART—cocaine- and amphetamine-regulated transcript; IGF1—insulin-like growth factor 1. Leptin has a direct anabolic effect on bone metabolism, however, it also modulates bone physiology indirectly by acting on the hypothalamus, acting on the sympathetic nervous system and by affecting the endocrine system. The direct action of leptin on the bone consists of osteoblast differentiation, and the enhanced synthesis of type I collagen and osteoprotegerin, thus increasing bone mineralization and inhibiting osteoclast activity and bone resorption. The effect of hormones on bone tissue is also relatively well known. It is known that leptin, by acting in the brainstem, regulates the function of the endocrine glands by affecting the pituitary gland and hormones produced by the ovary, parathyroid gland or adrenal cortex, which regulate bone metabolism. Still the most controversial is the role of leptin in the hypothalamus and the process of bone formation. The balance of the central and peripheral effects of leptin on bone metabolism remains unspecified, although recent studies have demonstrated that leptin’s direct effects on bone cells may override its actions via the central nervous system.

**Table 1 ijms-24-06840-t001:** Summary of origin, main functions and specific function in bone metabolism of the preeminent hormones included in this review.

Hormone	Structure	Origin of the Hormone	Main Functions	Specific Functions in Bone Metabolism	References
Kisspeptin	42.6 kDa protein 54 amino acids	Hypothalamus, placenta, right lobe of liver, nucleus accumbens, pituitary gland, duodenum	regulator of reproductive function, regulation of the onset of puberty, ovarian function, trophoblast invasion, fertility regulation, parturition, lactation, tumor suppression,	promotion of osteogenic differentiation of osteoblast progenitors inhibition of bone resorption increase in osteocalcin promotion of osteoblastogenesis	[18,20,21]
Neuropeptide Y	36 amino acid neuropeptide	ganglionic eminence, putamen, caudate nucleus, nucleus accumbens, amygdala, middle frontal gyrus, prefrontal cortex	stimulation of food intake, vasoconstriction, growth of fat tissue, anxiety reduction, decreased pain perception, circadian rhythm modification, reduction of voluntary alcohol intake, decrease in blood pressure	decreased runt-related transcription factor 2 (runx2) expression during osteogenic differentiation regulation of bone mass relating to the centrally perceived energy status of the body	[23,25,27]
Neuromedin U	25 amino acid peptide	hypothalamus, hippocampus, substantia nigra, spinal cord	contraction of smooth muscle, regulation of blood pressure, pain perception, appetite, hormone release	restriction of osteoblastic differentiation	[33,34]
CART	neuropeptide	hypothalamus, pituitary endocrine cells, adrenomedullary cells, islet somatostatin cells,	reward, feeding behaviors, stress, energy homeostasis	mediator of leptin regulation of bones decrease in number of osteoclasts	[38,39]
Endocannabinoids	lipid-based retrograde neurotransmitters	brain, immune cells, connective tissue	fertility, pre- and postnatal development, immune system, appetite, pain sensation, mood, and memory	stimulation of osteoclast activity increasing peak bone mass regulation of adipocyte and osteoblast differentiation of bone marrow stromal cells	[42,43]
Follicle- stimulating hormone	35.5 kDa glycoprotein heterodimer, two polypeptide units: alpha and beta	gonadotropes of the pituitary cells	stimulation of primordial germ cell maturation, induction of androgen-binding protein secretion by Sertoli cells in males, initiation of follicular growth in females	stimulation of osteoclast formation and function stimulation of bone resorption	[44,45]
Prolactin	23 kDa four-α-helix bundle protein, 199 amino acids	pituitary gland, stromal cell of endometrium, gastric mucosa, secondary oocyte, right coronary artery	milk production, enlargement of the mammary glands, maternal behavior, immune system regulation, regulation of hematopoiesis and angiogenesis	promotion of bone resorption stimulation of osteoclastogenesis inhibition of osteoblastogenesis	[47,48]
Adrenocorticotropic hormone	4540 Da 39 amino acids	anterior pituitary gland	stimulation of glucocorticoid steroid secretion, stress response	prevention of glucocorticoid-induced osteonecrosis stimulation of osteoblast proliferation	[49,50]
Thyroid-stimulating hormone	glycoprotein, consists of two subunits	anterior pituitary gland	regulation of thyroid endocrine function	improvement of bone microstructure prevention of osteoporosis	[51,52]
Growth hormone	22 kDa protein 191 amino acids	pituitary gland, liver	muscle growth, increases calcium retention, promotion of lipolysis, increased protein synthesis, immune system stimulation	protect against bone loss	[55]
Leptin	16 kDa protein 167 amino acids	adipose tissue, skeletal muscle, gastric mucosa, placenta, heart, mammary and salivary glands	food intake, non-shivering thermogenesis, hemostasis, angiogenesis, arterial pressure control, reproduction, neuroendocrine and immune functions	increase in bone mineralization increase in the activity of osteoblasts promotion of osteoblasts differentiation inhibition of bone resorption decreased osteoclast activity inhibition of osteoclast differentiation	[57,67,68,69,71]
Adiponectin	28 kDa protein 244 amino acids	Adipose cells, skeletal muscle, neurons	Insulin sensitization, inflammatory processes, cell proliferation, carcinogenesis	inhibition of osteoclasts differentiation suppression of bone resorption promotion of osteoblasts mineralization activity	[60,82,83,84,85,86]

## Data Availability

No new data were created or analyzed in this study. Data sharing is not applicable to this article.

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
