# Peer review of "Bones and Hormones: Interaction between Hormones of the Hypothalamus, Pituitary, Adipose Tissue and Bone"

_ijms, 2023, doi:10.3390/ijms24076840_

Round 1

Author Response

"Please see the attachment

Reviewer 2 Report

The aim of the reviewed work was to analyze and evaluate the current literature on bones and hormones: interaction between hormones of the hypothalamic, pituitary, and adipose tissues and bone. There are few publications in the available literature describing the issues of hormone effects on bone function in this approach. Understanding the hormonal, and functional biology of bone is important to better understand the bone as a highly dynamic structure with diverse functions. Developing a deeper understanding of bone tissue will certainly help in the development of new treatments for many bone diseases. The subject of review article is adequate to its content. The article is written in a clear and understandable way. The work is interesting, but the authors did not avoid some deficiencies. The work is interesting, but the authors did not avoid some deficiencies.

Recommendations

When using an abbreviation for the first time, expand it - - line 79 - KISS1

Line 293 -central nervous system (CNS) abbreviation has been used before

Line 203 the authors wrote - Prolactin receptors (PR) - the abbreviation PR refers to progesterone, not prolactin

It would be useful to standardize citations.

In some citations italics have been used

Some lines were used a citation after the author:

line 188 Li Sun et al. [32]; line 382- Sun et al ;

Some lines lack citations:

line-224 Li Sun et al; line -271 Calbet et al.;

In other lines, only the date is given instead of the citation:

line 322- Motyl et al (2012);

line 323- Gordeladze et al (2003);

line 325- Legiran et al (2012);

line 334- Takeda et al (2002);

line 346- Upadhyay et al (2015)

In references, some of the dates are bolded

Item 19 has been omitted from the citation

The subject of article is adequate to its content. The article is written in a clear and understandable way. In my opinion, the manuscript may be ready for publishing but after corrections.

Reviewer 3 Report

Niwczyc and colleagues reported an innovative and updated review of lesser known mechanisms of bone turn-over with points of future therapeutic perspectives.

Some comments and suggestions:

- pay attention to spaces throughout the text

- line 27 circuit and no circuitry

- line 40 very generalizing "hormone chenges and stressors"

- line 59 it may be interesting to indicate the age of attainment of peak bone mass

- I would not divide bone metabolism from other effects into separate paragraphs in either the kisspeptin chapter and  the next one

- line 91 put "directly" in the fine of the frase 

- names of all genes must be written in italics

- line 120 too many commas between " existing.....

- line 120 o study or authors

- line 131 amino acids that primarily act in the...

- line 134 I would remove "to a lessere extent" and "and i also expressed", line 138 " in turn", line 140 "at", line 142 "functions as diverse as"

- line 146-148 to clarify , difficult to understand

- line 161-162 repetitive with already said above

- line 179-180 use either all acronyms or all full names in the paragraph title

- line 182 i would remove " as they act"

- line 226 line low dose OF... 

- line 243 i would remove "controlling..... , line 261 "as it relates to energy regulation"

- line 266-268 to clarify , difficult to understand

- a lot of data on the effect of leptin in general with much less represented  part on the effects on bone metabolism which is the main subject of this review

- line 313-314 concept already said and repeated several times

- in paragraph n.8 reported the year of the scientific papers not respecting the format of this article and is different from other paragraphs of the article itself

- line 340 also ininitE 

- line 342 full name  acronym in brackets

- line 346-347 to clarify , difficult to understand

- line 352-353 use act and non acted, improve and non improved 

- line 368 formulate the sentence better

- in figure 1 it is not exactly correct to indicate estrogens etc. deriving from the pituitary, I would insert either tropine in the middle or figures of the glands that produce these hormones; fix graphic of the title of the figure

- line 381 I would replace treatment with practice

- line 387-391 mechanism already described above

- line 402 "chronic rheumatic inflammatory desases" never mentioned before, I would omit

-line 412 was shown

- line 421 omit of

- line 423 repetitive resolution to resolve

- line 423-424 to clarify , difficult to understand

- line 427 chemerin never mentioned before,add it also in the general part? 

- line 452 never mentioned before, no mention in the conclusions of what has not been described in the article 

- line 458 I would replace deepened with amplified

Round 2

Reviewer 1 Report

The authors have responded to all the points I raised in the previous review(s). I have no further issues.

Author Response

Dear Reviewer

Thank you for all your comments.